# Phytochemical Properties and In Vitro Biological Activities of Phenolic Compounds from Flower of *Clitoria ternatea* L.

**DOI:** 10.3390/molecules27196336

**Published:** 2022-09-26

**Authors:** Chao Li, Wei Tang, Shanglong Chen, Juping He, Xiaojing Li, Xucheng Zhu, Haimei Li, Yao Peng

**Affiliations:** 1College of Food and Bioengineering, Xuzhou University of Technology, Xuzhou 221018, China; 2College of Light Industry and Food Engineering, Nanjing Forestry University, Nanjing 210037, China; 3School of Life Sciences, Guangzhou University, Guangzhou 510006, China

**Keywords:** *Clitoria ternatea* L., phenolic compounds, phytochemical properties, antioxidant activities, enzyme inhibitory activities, antiproliferative activities

## Abstract

Phenolic compounds from the flower of *Clitoria ternatea* L. (PCFCTL) were extracted using a high-speed shearing extraction technique and purified by AB-8 macroporous resins, and the phytochemical composition of the purified phenolic compounds from the flower of *Clitoria ternatea* L. (PPCFCTL) was then analyzed. Subsequently, its bioactivities including antioxidant properties, enzyme inhibitory activities, and antiproliferative activities against several tumor cell lines were evaluated. Results indicated that the contents of total phenolics, flavonoids, flavonols, flavanols, and phenolic acids in PPCFCTL were increased by 3.29, 4.11, 2.74, 2.43, and 2.96-fold, respectively, compared with those before being purified by AB-8 macroporous resins. The results showed PPCFCTL have significant antioxidant ability (measured by reducing power, RP, and ferric reducing antioxidant power method, FRAP) and good DPPH, ABTS^+^, and superoxide anion radical scavenging activities. They can also significantly inhibit lipase, α-amylase, and α-glucosidase. In addition, morphological changes of HeLa, HepG2, and NCI-H460 tumor cells demonstrated the superior antitumor performance of PPCFCTL. However, the acetylcholinesterase inhibitory activity was relatively weak. These findings suggest that PPCFCTL have important potential as natural antioxidant, antilipidemic, anti-glycemic and antineoplastic agents in health-promoting foods.

## 1. Introduction

Phenolic compounds, existing in many kinds of plants, can be classified into two main groups: simple phenols and polyphenols [1]. The simple phenol group includes phenolic acids, while polyphenols include flavonoids (flavonols, flavanols, flavones, isoflavones), and lignans. Phenolic compounds are essential to human physiology because they play protective roles in human bodies, such as antioxidants [2], enzyme inhibitors [3], and antitumor and anti-inflammatory agents [4,5]. According to Zalba et al., oxidative stress is a physiological status where the production and metabolism of reactive oxygen species (ROS) are out of balance [6]. Oxidative stress is reported to be responsible for cancer [7], atherosclerosis [8], myocardial infarction [9], etc. An increasing number of studies have shown that most plant phenolic compounds have antioxidant properties with effects related to the number and position of phenolic hydroxyl groups and other special chemical structures. In addition, plant phenolic compounds are of great significance in inhibiting enzyme and anti-proliferation. Modern medical research has shown that many diseases that afflict people are closely related to the abnormal activities of various enzymes.

To control the activities of these enzymes so as to indirectly cure related diseases, scientists have carried out many studies on phenolic compounds. It has been reported that the polyphenolic-rich fraction of *Terminalia paniculata* bark could inhibit lipase activity [10]. Moreover, the polyphenols of five cultivars of young apple, including Short-Branch Fuji, Pink Lady, Jinshiji, Ruiyang, and Qinyang, reportedly have the ability to inhibit α-amylase activities, demonstrating a certain therapeutic effect in treating diabetes [11]. The polyphenol-rich extracts from jute leaves have been reported to exhibit a significant inhibitory effect on α-glucosidase activities, which may be helpful for the treatment of diabetes mellitus [12]. In addition, previous studies have also found that the cholinergic enzyme is linked to Alzheimer’s disease, and the inhibition of cholinergic enzymes such as acetylcholinesterase (AChE) and butyrylcholinesterase (BChE) has been recognized as an acceptable treatment for this disease [13]. Furthermore, the polyphenols extracted from lotus seed epicarp significantly inhibited the proliferation of HepG2 cells, a representative type of cancer cells [14].

*Clitoria ternatea* L., belonging to the Fabaceae family, is distributed in tropical and subtropical regions, including the Caribbean area, Central America, Africa, Southeast Asia, and India [15,16]. Its roots, stems, leaves, flowers, and seeds are commonly used in traditional medicine and diets. In combination with ginger powder, fixed oil, tannic acid, glucose, and bitter acid resin in the seeds can be powerful laxative agents. The seeds can also be used as food dye [17]. The roots have cooling, laxative, diuretic, anthelmintic, and anti-inflammatory properties which are of significance in the treatment of severe bronchitis, asthma, and hectic fever [18]. Stems are recommended for the treatment of snakebites and scorpion stings in India [19]. Leaves contain ester and resin glycosides [17], which are used in the treatment of several ailments including body aches, especially infections, and urogenital disorders. Leaves could also be utilized as anthelmintics and antidotes to insect stings [20]. Flowers are a good source of dietary anthocyanins and are used as natural blue colorants in a variety of foods, which have antimicrobial and anti-inflammatory activities [21,22,23]. However, not many reports have been published on the phytochemical properties or antioxidant, enzyme inhibitory, and antiproliferative activities of phenolic compounds in vitro from the flower of *Clitoria ternatea* L.

This study aimed to analyze the phytochemical properties of the purified phenolic compounds from the flower of *Clitoria ternatea* L. (PPCFCTL). We are the first to investigate their antioxidant capacity and enzyme inhibitory activity against lipase, alpha-amylase, alpha-glucosidase, and acetylcholinesterase, as well as their antiproliferative activity. We aimed to obtain a theoretical basis for the development of related products derived from PPCFCTL.

## 2. Results and Discussion

### 2.1. Phytochemical Analysis

AB-8 macroporous resin has been a common packing of the column in the purification of plant phenolics on account of its physicochemical stability, adsorption/desorption selectivity, and recyclability. The application of AB-8 macroporous resin in the purification of plant phenolic compounds has been reported in many previous studies [24,25,26]. In this research, AB-8 macroporous resin also demonstrated its outstanding performance in purifying the phenolic compounds from flower of *Clitoria ternatea* L. (PCFCTL)**.** In this study, the contents of total phenols, flavonoids, flavonols, flavanols, and phenolic acids were analyzed with UV–Vis spectrophotometry before and after purification, as shown in Table 1, where AB-8 macroporous resin performs well in PCFCTL purification. These significant increases in the purities of total phenolics, flavonoids, flavonols, flavanols, and phenolic acids indicated the high efficiency of AB-8 macroporous resin, which could be considered as an ideal purification media. The improvement of the purity of phenolic compounds is often of great significance for the evaluation of their biological activities [27].

### 2.2. Antioxidant Activities

DPPH is a kind of free-radical-generating substance which is widely used to monitor the free radical scavenging properties of different antioxidants [28]. As plotted in Figure 1A, there was a positive correlation between the DPPH radical scavenging activities of PPCFCTL and ascorbic acid and the concentrations. When the concentrations of PPCFCTL and ascorbic acid were in the range of 10 to 100 and 1 to 10 µg/mL, the scavenging rates were from 11.20 ± 0.37% to 59.40 ± 1.04% and from 8.58 ± 1.54% to 69.66 ± 0.53%, and their 50% scavenging concentration (SC_50_) values were 74.96 and 6.74 µg/mL, respectively. According to the corresponding SC_50_ values, the order of the DPPH radical scavenging activities was ascorbic acid > PPCFCTL.

The ABTS method is the most widely accepted colorimetric method to measure and characterize antioxidant capacity. One electron of an antioxidant combines with one electron of an ABTS^+^ radical, which can make the green color of the ABTS^+^ radical solution fade away [29]. As shown in Figure 1B, it was seen that the PPCFCTL (from 3 to 30 μg/mL) were able to dose-dependently inhibit the ABTS^+^ radical scavenging activities. The effects of PPCFCTL, which were expressed as percentages of inhibition, were from 23.38 ± 0.30% to 82.63 ± 0.85%. Ascorbic acid employed as a positive control exhibited notable ABTS^+^ radical scavenging activities from 8.21 ± 0.70% to 90.43 ± 0.28% as the concentrations varied from 0.9 to 9.0 μg/mL. The results demonstrated that the SC_50_ values of PPCFCTL and ascorbic acids were 9.90 and 4.48 μg/mL.

DNA molecules are vulnerable when exposed to superoxide anion radicals which can further generate active free radicals, including hydroxyl radical, singlet oxygen, and hydrogen peroxide [30]. As shown in Figure 1C, the scavenging rate of PPCFCTL on superoxide anion radicals rose upon the increase in the concentrations. The scavenging rates of PPCFCTL and ascorbic acid were 13.96 ± 1.97% and 27.60 ± 1.89% when the concentrations were 8 and 2 μg/mL, respectively. When the concentrations increased to 80 and 20 μg/mL, the rates rose to 77.21 ± 2.15% and 95.90 ± 1.42%. The SC_50_ value of PPCFCTL was 39.24 μg/mL, while the value of ascorbic acid was 4.24 μg/mL.

The reduction in Fe^3+^ indicates an electron donor. Due to their reducing power, phenolic compounds act as electron and/or hydrogen donors to scavenge free radicals in vivo [31]. The activity of PPCFCTL was concentration-dependent as described in Figure 1D. When the concentrations varied from 12 to 120 μg/mL, the reducing power increased from 0.095 ± 0.006 to 0.590 ± 0.015. The EC_50_ values of PPCFCTL and ascorbic acid were 86.77 and 6.78 µg/mL, which proved that PPCFCTL had notable reducing power.

The ferric ion reducing antioxidant power (FRAP) assay is based on an electron transfer reaction where ferric salt is used as an antioxidant [32]. As depicted in Figure 1E, PPCFCTL showed dose-dependent FRAP within the tested concentration range. At the concentrations of 4–40 μg/mL, the FRAP of PPCFCTL increased from 0.136 ± 0.010 to 0.710 ± 0.016. The EC_50_ value of PPCFCTL was found to be 25.90 μg/mL, which was lower than that of ascorbic acid (2.98 μg/mL).

### 2.3. Enzyme Inhibitory Activities

Pancreatic lipase, a fat decomposing enzyme, which is secreted by the pancreas, is vital in the digestion of triglycerides [33]. Therefore, the application of lipase inhibitors may inhibit the degradation of fat catalyzed by lipase, which reduces the absorption of fat by the human body. Figure 2A shows that the inhibitory effect of PPCFCTL on lipase rose as the concentrations of PPCFCTL increased. The lipase inhibitory rates of PPCFCTL were from 17.71 ± 1.58% to 82.89 ± 3.68%, when the concentrations ranged from 0.075 to 0.60 mg/mL. The 50% inhibitory concentration (IC_50_) value of PPCFCTL was 0.28 mg/mL, which was higher than that of orlistat (0.16 mg/mL). However, it is still an effective lipase inhibitor.

α-amylase, the inhibition of which is considered a strategy in the treatment of type 2 diabetes, is an enzyme that plays a key role in the digestion of starch [34]. As a result, α-amylase has also been recognized as a therapeutic target for the modulation of postprandial hyperglycemia. It can be seen from Figure 2B that there was a gradual increase in α-amylase inhibitory effect with increasing concentrations of PPCFCTL. At the concentrations of 0.45–3.60 mg/mL, the α-amylase inhibitory rates varied from 13.67 ± 2.63% to 77.72 ± 1.92%. The IC_50_ value of PPCFCTL was 1.70 mg/mL, while the IC_50_ value of acarbose was 2.76 mg/mL. From the above results, it could be found that PPCFCTL had stronger α-amylase inhibitory activities.

α-glucosidases in the brush border of the small intestine are essential for the process of degradation from more complex carbohydrates to glucose, which inhibits postprandial glucose peaks, thereby leading to decreased post-load insulin levels [35]. Thus, α-glucosidase is an important target enzyme for the treatment of type 2 diabetes in humans [36]. It could be concluded from Figure 2C that both PPCFCTL and ascorbic acid inhibited α-glucosidase activities in a concentration-dependent manner at the tested concentrations. The IC_50_ values for the inhibition of α-glucosidase activities by PPCFCTL and acarbose were found to be 1.04 and 0.94 mg/mL, respectively. Based on the obtained results, PPCFCTL had a potent inhibitory effect against α-glucosidases.

Acetylcholinesterase is a serine hydrolase [37]. Its principal biological function is to terminate the impulse transmissions at cholinergic synapses within the nervous system by rapidly hydrolyzing the neurotransmitter acetylcholine [38]. Inhibition of acetylcholinesterase is an important strategy for the treatment of Alzheimer’s disease and other cholinergic transmission deficiency diseases [39]. It can be clearly seen in Figure 2D that the acetylcholinesterase inhibitory effects of both PPCFCTL and galantamine were well correlated with the concentrations, and the acetylcholinesterase inhibitory effect of galantamine was significantly higher than that of PPCFCTL. At the concentration of 0.18 and 1.44 mg/mL, the acetylcholinesterase inhibitory rates of PPCFCTL increased from 14.25 ± 2.24% to 94.85 ± 4.38%; similarly, at the concentration of 8 × 10^−6^ and 6.4 × 10^−5^ mg/mL, the acetylcholinesterase inhibitory rates of galantamine increased from 38.56 ± 1.83% to 77.00 ± 1.33%. The IC_50_ values of PPCFCTL and galantamine were 0.47 and 1.6 × 10^−5^ mg/mL, respectively. This result suggested that the PPCFCTL were weak acetylcholinesterase inhibitors.

### 2.4. Antiproliferative Activities

In vitro evaluation of anticancer properties using cancer cell lines is an important tool in discovering new anticancer drugs with high specificity [40]. MTT and other tetrazolium haline-based detection methods are among the most popular techniques for quantitative assessment of cell proliferation, viability, and cytotoxicity due to their low cost, high efficiency, and simplicity [41]. The typical MTT method is the colorimetric determination on a microtiter plate and the absorbance at the end of the measurement. Tetrazolium acts as an indicator of intracellular reduction potential, which in turn indicates the overall state and viability of cells. The enzymolysis of and reduction in tetrazolamide by the cell dehydrogenase and reducing agent resulted in violet blue and water-insoluble formazan products [42]. The activities of PPCFCTL on the proliferation of HeLa, HepG2, and NCI-H460 cells are illustrated in Figure 3A–C. It was obvious that PPCFCTL inhibited the proliferation of all the three cell lines in a dose-dependent manner. When the concentration increased from 150 to 900 μg/mL, the inhibition rates for 24 h of PPCFCTL treatment ranged from 13.98 ± 4.58% to 81.88 ± 1.35%, 0.66 ± 1.51% to 18.48 ± 3.18%, and 27.71 ± 1.40% to 82.77 ± 3.24%, and for 48 h ranged from 17.82 ± 5.31% to 87.54 ± 2.34%, 53.97 ± 3.70% to 57.06 ± 3.23%, and 36.39 ± 1.17% to 94.51 ± 0.41%, for the HeLa, HepG2, and NCI-H460 cells, respectively. Thus, the effect of 48 h treatment was generally better than that of 24 h treatment. The experimental results were similar to those of Chen et al. [43].

### 2.5. Cellular Morphology

Cell morphology is a common index reflecting the physiological and growth state of cells. The observation of cell morphology can be used to judge the physiological and growth state of cells, so as to further reflect the effect of drugs on cell growth [44]. Morphological changes of HeLa, HepG2, and NCI-H460 cells treated and untreated with different concentrations of PPCFCTL were observed after 24 and 48 h. As shown in Figure 4, Figure 5 and Figure 6, according to the model control group, it could be seen that the cells of HeLa, HepG2, and NCIH460 all presented irregular shapes and grew adherent to the inner wall of the culture plate. Especially after 48 h of culture, the cells were tightly connected and increased significantly in number. However, compared with the model control group, after PPCFCTL (750 and 900 μg/mL) or 5-fluorouracil (5-FU) treatment for 24 and 48 h, the number of adherent cells decreased significantly, and the cell morphology changed from an irregular shape to a round shape. At the same time, it could also be seen that the 900 μg/mL PPCFCTL treatment was more effective than that of 750 μg/mL. Collectively, these results suggested that PPCFCTL effectively inhibit the proliferation of HeLa, HepG2, and NCI-H460 cells in vitro. The result of the assay was similar to that of the literature [45].

## 3. Materials and Methods

### 3.1. Plant Material

*Clitoria ternatea* L. was purchased from Guangxi Qinzhou BaiCaoYuan Biotechnology Co., Ltd. (Qinzhou, Guangxi, China). Its flower was taxonomically authenticated and deposited with voucher number no. 20180322-2 at the herbarium of Food Department, Xuzhou University of Technology, Xuzhou. It was dried in an oven (GZX-9070MBE, Boxun, Shanghai, China) with air circulation at 60 °C, and the dried material was ground to fine, homogeneous powder using a grinder (WKX-160, Jingcheng, Qingzhou, Shandong, China), sieved through a 60-mesh sieve, and kept in a sealed plastic bag at −20 °C prior to use.

### 3.2. Chemicals and Reagents

Gallic acid (≥98%), rutin (≥98%), catechin (≥98%), pyrogallol (≥98%), ascorbic acid, *p*-nitrophenyl palmitate (*p*-NPP), orlistat, 3,5-dinitrosalicylic acid (DNS), *p*-nitrophenyl-α-D-glucopyranoside (*p*-NPG), 5,5′-dithiobis(2-nitrobenzoic acid) (DTNB), and acetylthiocholine iodide (AChI) were purchased from Hefei Bomei Biotechnology Co., Ltd. (Hefei, Anhui, China). 3-(4,5-Dimethylthiazol-2-yl)-2,5-diphenyltetrazolium bromide (MTT, M5655, ≥97.5%) was purchased from Sigma-Aldrich Co., Ltd. (St. Louis, MO, USA). Folin–Ciocalteau reagent was purchased from Shanghai Jinsui Biotechnology Co., Ltd. (Shanghai, China). *p*-(dimethylamino)cinnamaldehyde (*p*-DMACA) and galanthamine were purchased from Shanghai Yuanye Biotechnology Co., Ltd. (Shanghai, China). The human cervical carcinoma cell line HeLa, the human liver cancer cell line HepG2, and human large cell lung cancer cell line NCI-H460 were obtained from the Cell bank of typical culture preservation Committee of the Chinese Academy of Sciences. Trypsin–EDTA solution and 5-FU (F6627, ≥99%) were purchased from Beyotime Biotechnology Co., Ltd. (Shanghai, China). All other chemicals used were of analytical grade and procured from Sinopharm Chemical Reagent Co., Ltd. (Shanghai, China). AB-8 macroporous resin was purchased from Anhui Sanxing Resin Technology Co., Ltd. (Bengbu, Anhui, China).

### 3.3. Preparation of PPCFCTL

#### 3.3.1. Extraction of PCFCTL

The flower powder (30.0 g) was weighed and transferred into the extraction container (2.0 L). Thereafter, 750 mL of 80% ethanol (*v*/*v*) was added and emulsified with a high-speed shearing machine (ZHBE-50, Zhijing, Zhengzhou, China). The extraction process lasted 10 times, 200 s each time at a machine voltage of 100 V. The crude extracts were then collected, filtered, concentrated, and lyophilized for further purification.

#### 3.3.2. Purification of PCFCTL

PCFCTL purification was implemented as previously reported with applicable modifications [24]. First, AB-8 macroporous resins were put into a chromatographic column (2.6 × 60 cm) using the wet method, and the final bed volume (BV) was 320 mL. Then, a dynamic adsorption test was conducted on the column. The sample solution consisted of 2 g crude extract and 500 mL deionized water, and the pH was adjusted with 1 mol/L HCl to 4.6. The solution was added to the column with a constant flow pump at a flow rate of 1 BV/h. After that, the dynamic desorption test was performed. The column was eluted with distilled water until the eluent became transparent. Later, 3.5 BV of 80% ethanol was used to rinse the column at 0.5 BV/h. Finally, the eluted solution was collected, concentrated, and lyophilized. The lyophilized product after purification is called PPCFCTL. The schematic diagram of the extraction and separation of PCFCTL from *Clitoria ternatea* L. is shown in Figure 7.

### 3.4. Phytochemical Detemination

Total phenolic content was measured using the reported method with minor modifications [46]. According to the calibration curve of rutin, the phenolic content was determined as mg GAE/g (freeze-dried sample). GAE stands for gallic acid equivalent.

Total flavonoid content was measured using the reported method with minor modifications [47]. According to the calibration curve of rutin, the flavonoid content was determined as mg RE/g mg (freeze-dried sample). RE stands for rutin equivalent.

Total flavonol content was measured using the reported method with minor modifications [48]. According to the calibration curve of rutin, the flavonol content was determined as mg RE/g (freeze-dried sample). RE stands for rutin equivalent.

Total flavanol content was measured using the reported method with minor modifications [49]. According to the calibration curve of catechin, the flavanol content was determined as mg CE/g (freeze-dried sample). CE stands for catechin equivalent.

Total phenolic acid content was measured using the reported method with minor modifications [50]. According to the calibration curve of caffeic acid, the phenolic acid content was determined as mg CAE/g (freeze-dried sample). CAE stands for caffeic acid equivalent.

### 3.5. Antioxidant Assays

#### 3.5.1. DPPH Radical Scavenging Assay

The DPPH radical scavenging activities were measured using the reported method with minor modifications [51]. The process was as follows: 50 µL of varying concentrations of PPCFCTL was added to 150 µL of 0.15 mM DPPH (in ethanol) in 96-well plates. Then, the mixed solution was incubated in the dark at 37 °C for 1.0 h; afterward, the absorbance of the solution at 517 nm was measured using a microplate reader (Synergy H1, Bio-Tek, Winooski, VT, USA). Ascorbic acid served as a positive control. The scavenging rate was calculated using the formula below (*A_s_* stands for the absorbance of the sample group and *A_c_* for the blank control group.):(1)$$Inhibition\;rate\;\left(\%\right)=\left(1-\frac{A_{s}}{A_{c}}\right)\times 100$$

#### 3.5.2. ABTS^+^ Radical Scavenging Assay

The ABTS^+^ radical scavenging activities were measured using the reported method with minor modifications [51]. The process was as follows: 5 mL of 7 mM ABTS and 88 μL of 140 mM potassium persulphate were mixed in 96-well plates to generate ABTS^+^ radicals. Then, the mixture was left to stand for 12–16 h in the dark at room temperature which adjusted the absorbance at 734 nm of the solution to 0.85 ± 0.02. Afterward, 40 μL of varying concentrations of PPCFCTL was mixed with 160 μL of ABTS solution in 96-well plates; the mixture was incubated in the dark at 37 °C for 1 h. Finally, the absorbance of the solution after incubation was measured using a microplate reader. Ascorbic acid was employed as a positive control. The ABTS^+^ radical scavenging rate was calculated using Equation (1).

#### 3.5.3. Superoxide Anion Radical Scavenging Assay

The superoxide anion radical scavenging activities were measured using the reported method with minor modifications [51]. The process was as follows: 20 µL of varying concentrations of PPCFCTL was added to 100 µL of Tris-HCl buffer (50 mM, pH 8.20) in 96-well plates. Then, the mixed solution was incubated in the dark at 37 °C for 20 min. Afterward, 8 µL of pyrogallol, namely 3 mM of pyrogallol in 10 mM of HCl, which was pre-incubated in the dark at 37 °C for 5 min, was injected into the plates, and the mixture was incubated in the dark at 37 °C for 5 min. Soon after that, 32 µL of 1 M HCl was added to terminate the reactions. Finally, the absorbance at 320 nm of the mixed solution was measured using a microplate reader; ascorbic acid was employed as a positive control. The superoxide anion radical scavenging rate was calculated using the formula below (*A_s_* stands for the absorbance of the sample group, *A_sb_* for the sample background group, and *A_c_* for the blank control group):(2)$$Scavenging\;rate\;\left(\%\right)=\left(1-\frac{A_{s}-A_{sb}}{A_{c}}\right)\times 100$$

#### 3.5.4. Reducing Power (RP) Assay

The reducing power of PPCFCTL was measured using the reported method with minor modifications [51]. The process was as follows: 10 µL of varying concentrations of PPCFCTL was added to the mixed solution of 25 µL of 0.2 M phosphate buffer (PBS, pH 6.6) and 25 µL of 1% (*w*/*v*) potassium ferricyanide in 96-well plates. Then, the mixed solution was incubated in the dark at 37 °C for 30 min. Then, 25 µL of 10% (*w*/*v*) trichloroacetic acid was injected into the plates to terminate the reactions. Afterward, 17 µL of 0.1% (*w*/*v*) ferric chloride and 85 µL of distilled water were added to the mixture. Finally, the absorbance of the mixed solution at 700 nm was measured using a microplate reader; ascorbic acid was employed as a positive control. The reducing power effect of PPCFCTL was calculated using the formula below (*A_rp_* stands for the reducing power, while *A_s_* stands for the absorbance of the sample group and *A_c_* for the blank control group):(3)$$A_{rp}=\left(A_{s}-A_{c}\right)$$

#### 3.5.5. Ferric Reducing Antioxidant Power (FRAP) Assay

The ferric reducing antioxidant power of PPCFCTL was measured using the reported method with minor modifications [51]. The process was as follows: 10 mM TPTZ (in 40 mM HCl), 20 mM ferric chloride, and acetate buffer (0.3 M, pH 3.6) were mixed in the proportion of 1:1:10 (*v*/*v*/*v*) to produce the FRAP reagent which was prepared right before use. Then, 15 µL of varying concentrations of PPCFCTL was added to 185 µL of FRAP reagent in 96-well plates. Afterward, the mixed solution was incubated in the dark at 37 °C for 10 min. Finally, the absorbance of the solution at 593 nm was measured using a microplate reader; ascorbic acid was employed as a positive control. *A_frap_* stands for the ferric reducing antioxidant power and was calculated using Equation (3).

### 3.6. Enzyme Inhibitory Assays

#### 3.6.1. Lipase Inhibitory Assay

The inhibitory effect of PPCFCTL on lipase was measured using the reported method with minor modifications [52]. The experiment procedure, concentration and dose of the reagents, incubation conditions (37 °C, 10 or 20 min), and detection method (microplate reader, 405 nm) were the same as those of the reference. The 405 nm absorbance of the solution indicated the inhibitory rate of PPCFCTL on lipase (orlistat as a positive control), which was determined using the equation below. (*A_s_* stands for the absorbance of the sample group, *A_sb_* for the sample background group, *A_n_* for the negative control group, and *A_nb_* for the negative background control group): (4)$$Inhibition\;rate\;\left(\%\right)=\left(1-\frac{A_{s}-A_{sb}}{A_{n}-A_{nb}}\right)\times 100$$

#### 3.6.2. α-Amylase Inhibitory Assay

The inhibitory effect of PPCFCTL on α-amylase was measured using the reported method with minor modifications [53]. The experiment procedure, concentration and dose of the reagents, incubation conditions (37 or 100 °C, 5 or 20 min), and detection method (microplate reader, 540 nm) were the same as those of the reference. The 540 nm absorbance of the solutions indicated the inhibitory rate of PPCFCTL on α-amylase which was determined by Equation (4).

#### 3.6.3. α-Glucosidase Inhibitory Assay

The inhibitory effect of PPCFCTL on α-glucosidase was measured using the reported method with minor modifications [53]. The experiment procedure, concentration and dose of the reagents, incubation conditions (37 °C, 10 or 5 min), and detection method (microplate reader, 405 nm) were the same as those of the reference. The 405 nm absorbance of the solution indicated the inhibitory rate of PPCFCTL on α-glucosidase (acarbose as a positive control) which was determined by Equation (4).

#### 3.6.4. Acetylcholinesterase Inhibitory Assay

The inhibitory effect of PPCFCTL on acetylcholinesterase was measured using the reported method with minor modifications [53]. The experiment procedure, concentration and dose of the reagents, incubation conditions (37 °C; 5, 15, or 20 min), and detection method (microplate reader, 405 nm) were the same as those of the reference. The 405 nm absorbance of the final solution indicated the inhibitory rate of PPCFCTL on acetylcholinesterase (galantamine as a positive control) which was determined by Equation (4).

### 3.7. Antiproliferative Assays

#### 3.7.1. Cell Culture

The HeLa, HepG2, and NCI-H460 cells were cultivated in RPMI-1640. Heat-inactivated FBS (10%), 100 units/mL penicillin, and 100 μg/mL streptomycin which was stored in 37 °C humidified air with 5% CO_2_ were provided as a supplement, and they were from the Jiangsu Food Safety Biochip Testing Technology Engineering Laboratory, Xuzhou University of Technology, Xuzhou, Jiangsu, China.

#### 3.7.2. Cell Viability Assay Using MTT Method

The protocol of antiproliferative assays was adapted from the one reported by Guo et al. [54]. The process was as follows: the cells in the logarithmic growth period at 1 × 10^5^ cells/mL were prepared. Then, 100 μL of the cells and the samples were mixed to achieve the final concentrations of 0, 14.06, 28.12, 56.25, 112.50, 225.00, 450.00, and 900.00 µg/mL. Afterward, the mixture was shaken slightly for 1 min and incubated in a humidified atmosphere of 5% CO_2_ at 37 °C for 24 and 48 h; then, 20 μL of MTT (5 mg/mL, in PBS) was added, and the solution was incubated for another 4 h in an incubator with humidified air including 5% CO_2_ at 37 °C. After the supernatant was discarded, the formazan precipitates were dissolved in 150 μL DMSO after another 10 min of incubation at room temperature in the dark. The 490 nm absorbance indicated the inhibitory rate of PPCFCTL on cells (5-FU as a positive control), which was calculated using the formula below (*A_s_* stands for the absorbance of the sample group, *A_n_* for the negative control group, and *A_b_* for the blank control group): (5)$$Inhibition\;rate\;\left(\%\right)=\left(1-\frac{A_{s}-A_{b}}{A_{n}-A_{b}}\right)\times 100$$

#### 3.7.3. Cell Morphology

The morphological changes of HeLa, HepG2, and NCI-H460 cells were observed under an optical inverted microscope (TS100-F, Nikon, Chiyoda District, Tokyo Metropolitan, Japan) at ×100 magnification after treatment with 0, 750, 900 μg/mL of PPCFCTL and 150 μg/mL of 5-FU for 24 and 48 h.

### 3.8. Statistical Analysis

All experiments were performed in triplicate, and the results were reported in the form of mean ± standard deviation. The analysis was performed with using SPSS V18.0 software (IBM Co., Armonk, NY, USA), and the figures were plotted using Origin V9.1 software (Origin Lab Co., Northampton, MA, USA).

## 4. Conclusions

By the comparison of the contents of total phenolics, flavonoids, flavonols, flavanols, and phenolic acids in PCFCTL and PPCFCTL, conclusions can be drawn that AB-8 macroporous resin can effectively purify PPCFCTL. In this study, we provide the first in-depth assessment of the antioxidant, enzyme inhibition, and antiproliferative properties of PPCFCTL. The study revealed that PPCFCTL may be developed as naturally potential antioxidants or antilipidemic, antidiabetic, or antineoplastic drugs in the nutraceutical and pharmaceutical industries. However, their mechanisms of antilipidemic, antidiabetic, and antineoplastic activities in vivo need further research. The experimental results showed that PPCFCTL were not effective acetylcholinesterase inhibitors.

## Figures and Tables

**Figure 1 molecules-27-06336-f001:**
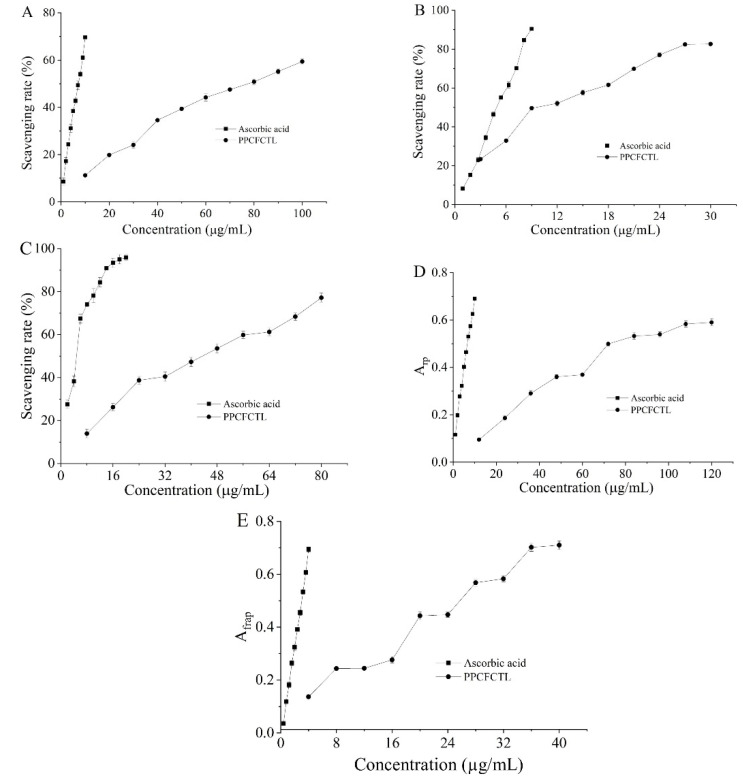
Antioxidant activities of PPCFCTL: (**A**) DPPH radical scavenging activities, (**B**) ABTS + radical scavenging activities, (**C**) superoxide anion radical scavenging activities, (**D**) reducing power activities, (**E**) ferric reducing antioxidant power activities. Data represent means of three independent experiments (mean ± SD).

**Figure 2 molecules-27-06336-f002:**
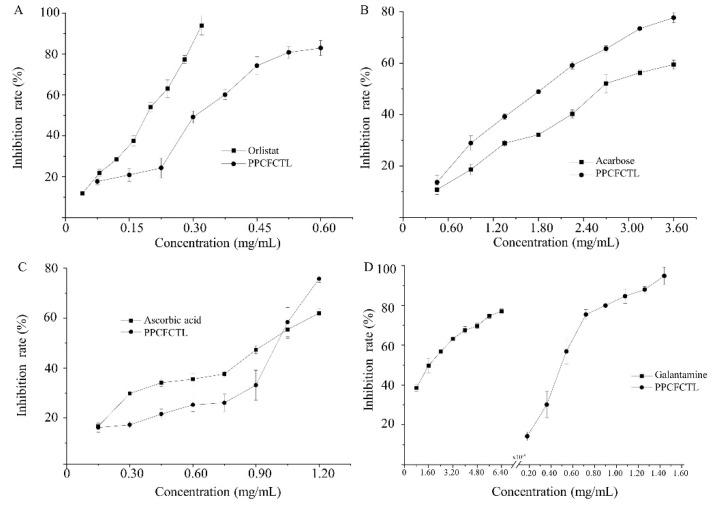
Enzyme inhibitory activities of PPCFCTL: (**A**) lipase inhibitory activities, (**B**) α-amylase inhibitory activities, (**C**) α-glucosidase inhibitor activities, (**D**) acetylcholinesterase inhibitory activities. Data represent means of three independent experiments (mean ± SD).

**Figure 3 molecules-27-06336-f003:**
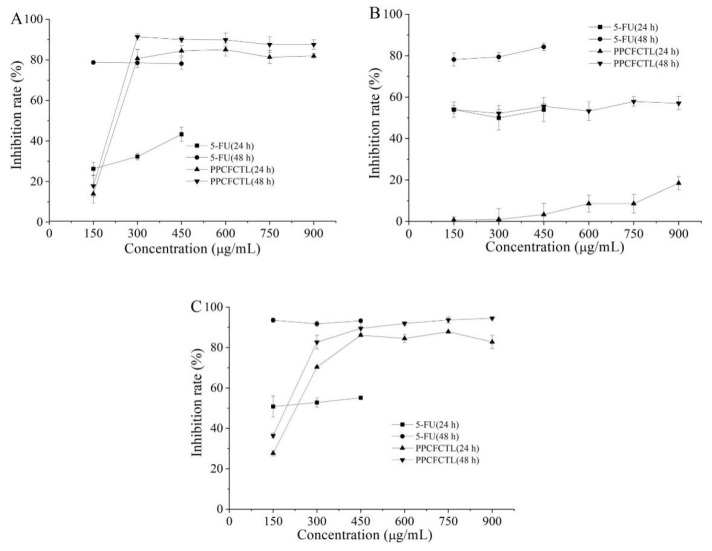
Antiproliferative activities of PPCFCTL: (**A**) Human cervical cancer HeLa cell, (**B**) Human hepatoma HepG2 cell, (**C**) Human lung cancer NCI-H460 cell. Data represent means of three independent experiments (mean ± SD).

**Figure 4 molecules-27-06336-f004:**
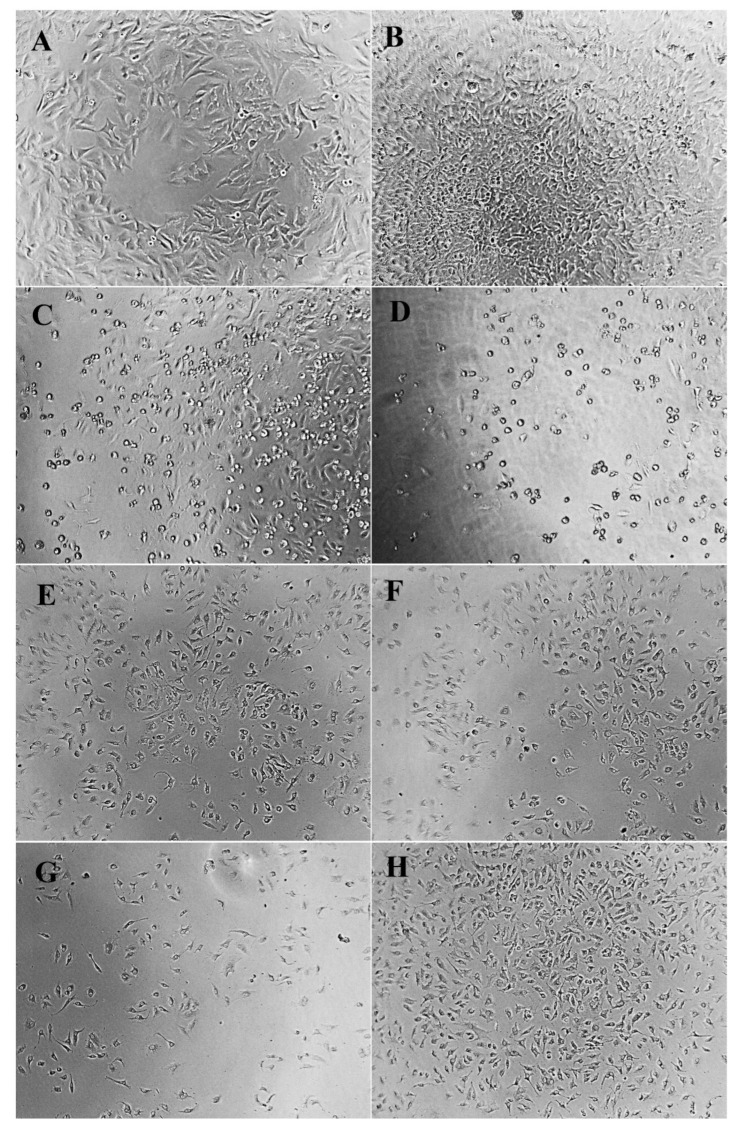
Effect of PPCFCTL on morphology of HeLa cells. Negative control group ((**A**) 24 and (**B**) 48 h), 150 µg/mL 5-FU treatment group ((**C**) 24 and (**D**) 48 h), 750 µg/mL PPCFCTL treatment group ((**E**) 24 and (**F**) 48 h), 900 µg/mL PPCFCTL treatment group ((**G**) 24 and (**H**) 48 h).

**Figure 5 molecules-27-06336-f005:**
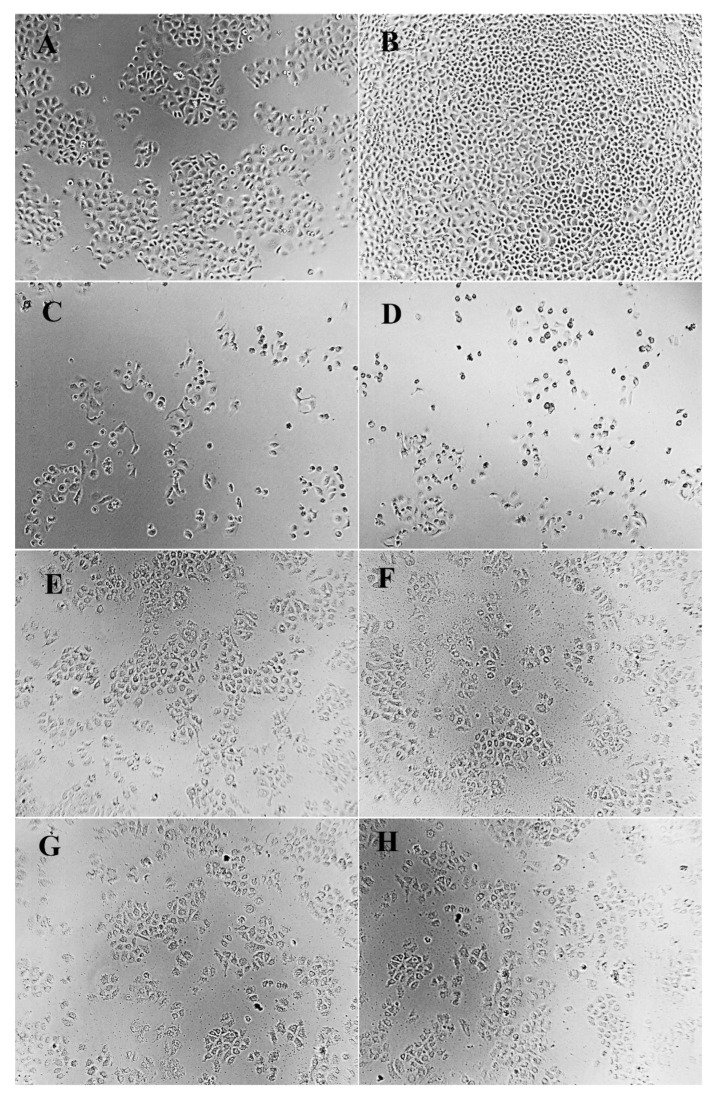
Effect of PPCFCTL on morphology of HepG2 cells. Negative control group ((**A**) 24 and (**B**) 48 h), 150 µg/mL 5-FU treatment group ((**C**) 24 and (**D**) 48 h), 750 µg/mL PPCFCTL treatment group ((**E**) 24 and (**F**) 48 h), 900 µg/mL PPCFCTL treatment group ((**G**) 24 and (**H**) 48 h).

**Figure 6 molecules-27-06336-f006:**
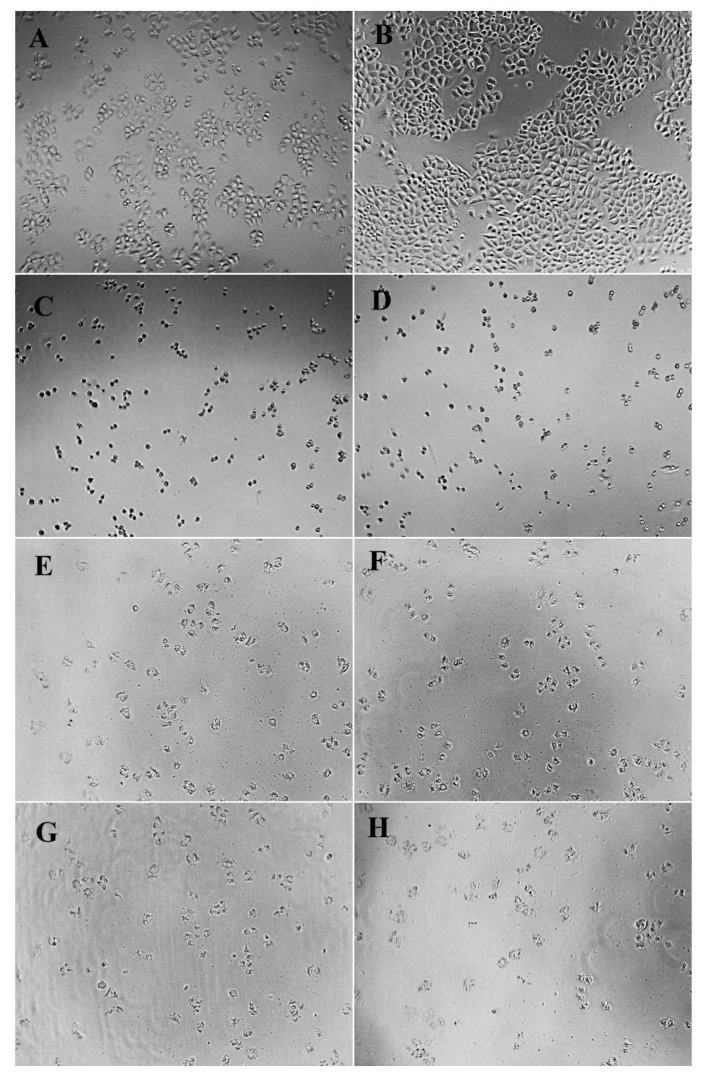
Effect of PPCFCTL on morphology of NCIH460 cells. Negative control group ((**A**) 24 and (**B**) 48 h), 150 µg/mL 5-FU treatment group ((**C**) 24 and (**D**) 48 h), 750 µg/mL PPCFCTL treatment group ((**E**) 24 and (**F**) 48 h), 900 µg/mL PPCFCTL treatment group ((**G**) 24 and (**H**) 48 h).

**Figure 7 molecules-27-06336-f007:**
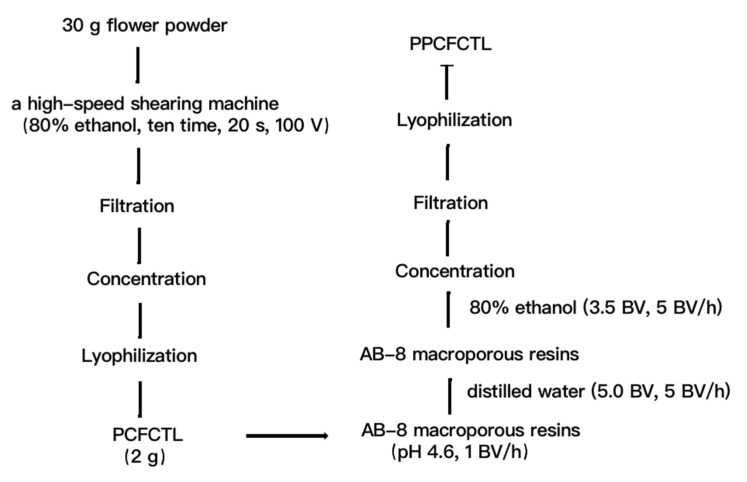
The schematic procedures for the extraction and separation of PCFCTL from *Clitoria ternatea* L.

**Table 1 molecules-27-06336-t001:** Phytochemical constituents of PCFCTL and PPCFCTL.

Phenolic Compounds	PCFCTL	PPCFCTL
Total phenolics (mg GAE/g)	55.24 ± 0.68 ^a^	236.78 ± 0.35 ^b^
Flavonoids (mg RE/g)	33.48 ± 0.44 ^a^	171.22 ± 0.91 ^b^
Flavonols (mg RE/g)	52.96 ± 0.40 ^a^	197.83 ± 1.69 ^b^
Flavanols (mg CE/g)	0.43 ± 0.02 ^a^	1.48 ± 0.05 ^b^
Phenolic acid (mg CAE/g)	14.83 ± 0.23 ^a^	60.04 ± 1.17 ^b^

Data are presented as the mean ± SD of three independent replicates. Means within a column followed by the same letter are not significantly different (*p* < 0.05).

## Data Availability

Not applicable.

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
