# Peer review of "Phytochemical Properties and In Vitro Biological Activities of Phenolic Compounds from Flower of Clitoria ternatea L."

_molecules, 2022, doi:10.3390/molecules27196336_

Round 1

Reviewer 1 Report

The paper analyzed the phytochemical profile from the flower of Clitoria ternatea L. and several biological activities were assayed.

The subject is important. However, for the improvement of the manuscript, please address the following suggestions:

The title seems too long; a more focused title, such as “Phytochemical properties and in vitro biological activities of ...” can have a higher impact.

English should be revised; there are small mistakes.

Clitoria ternatea should be written in italics throughout.

Acronyms/Abbreviations should be defined the first time they appear in each of three sections: the abstract; the main text; the first figure or table (when defined for the first time, the acronym/abbreviation should be added in parentheses (see Line 91: PCFC).

Figures should be in color (if possible).

In the text, reference numbers should be placed in square brackets [ ], and placed before the punctuation.

All Figures and Tables should be inserted into the main text close to their first citation and must be numbered following their number of appearance (correct the numbering for all Figures).

All Figures and Tables should have a short explanatory title and caption (please explain superscript a and b in Table 1).

When figures are mentioned in text, I suggest using capital letters, such as Fig. 2(A), Fig. 3(B), Fig. 4(A-C). Same for the captions.

Abstract: try not to use words like “outstanding, considerable, excellent”, instead use “good, significant, important”

Lines 35-36: Simple phenol group includes phenolic acids, while polyphenols include flavonoids (flavonols, flavanols, flavones, isoflavones), lignans, etc.” instead of “The former includes phenolic acids, etc, and the latter include flavonoids (flavonols, flavanols, flavones, isoflavones, etc), lignans, etc.

Line 38: after “antioxidants”, please add “anti-inflammatory” (doi: 10.3390/antiox11071412), knowing that inflammation is a key risk factor in many diseases 

Line 50: (TPPF)” can be deleted

Line 64: “etc.” can be deleted

Line 72: “Leaves could also be used as...”

Line 84: It should be Results and Discussion

Line 85: Presentation and discussion of the results are needed for subchapter 2.1. I suggest adding the following idea: “Despite the favorable in vitro potential of any compound or extract, the safety level and the beneficial outcomes could only be determined through in vivo toxicological studies (Vedeanu et al. Doi: 10.1071/EN19249)

Lines 97-100: pay attention at PCFC and PPCFC! It is “Table 1. Phytochemical constituents of phenolic compounds from flower of Clitoria ternatea L(PCFC) and purified phenolic compounds extracted from flower of Clitoria ternatea L. (PPCFC).” The 2nd column should be PCFC, and the 3rd column PPCFC!

Line 101: Discussion for subchapter 2.2. 

Line 106: “10 - 100 and 1 - 10 μg/mL, respectively,

Line 107: “8.58 ± 1.54% to 69.66 ± 0.53%, respectively

Line 125: based on the figure, “77.19 ± 2.15% is not correct!

Line 127: 27.60 ± 1.89%” is not correct!

Line 127: “...95.90 ± 1.42%, respectively.

Lines 141, 142: what is “PPCFCTL”? 

Line 153: “0.075 to 0.60 mg/mL” instead of “7.5×10-2 to 0.60 mg/mL 

Line 171: “77.72 ± 161 1.92%” is not correct!

Lines 187, 188: what is “PPCFCTL”? Revise Figures 3(A-D)! 3(a) is not pancreatic lipase inhibitory activities”! Fig. 3(D) has no galantamine curve!

Lines 210, 211: what is “5-FU” and “PPCFCTL”?

Lines 232, 236, 240: what is “PPCCMR”?

Lines 288-291: Is it extraction and separation of PCFC or PPCFC?

Lines 381-386: what was the positive control?

Reviewer 2 Report

The work performed concerns the search for new compounds of chemopreventive (there is no information about this in the paper and this is what we call the activity studied), anti-free radical and anti-diabetic nature. These are important problems resulting from the development of civilisation and modern lifestyle. Unfortunately, the level of some of the research presented in the paper is low.

The paper should improve:

1. The quality of Figures 5, 6 and 7. Improve contrast, add scale. For a better analysis of the images, I recommend adding a higher magnification photograph as well.

2. Point 3.7.3 photos do not look like 400x magnification

3. The work should be completed with the results obtained on normal cells. Antitumour activity is not only assessed for tumour cells.

4. For the MTT assay we evaluate the effect on cellular vitality on proliferation. In addition, when testing anti-free radical compounds, the MTT test often gives false-negative results. The work should be supplemented with an additional test to confirm the MTT test results. In addition, for the evaluation of anti-cancer activity, we will always use a control compound in the study for comparison.

Round 2

Reviewer 1 Report

The authors addressed the suggestions and the manuscript has improved. I just have two small suggestions:

Phenolic compounds from the flower of Clitoria ternatea L. (PCFCTL) and Purified phenolic compounds from the flower of Clitoria ternatea L. (PPCFCTLshould be defined only once in Abstract and text; then the abbreviation should be used.

Please check reference #27 (the names should be reversed - now the first names appear) 

Author Response

Response to Reviewer 1 Comments:

Point 1: Phenolic compounds from the flower of Clitoria ternatea L. (PCFCTL) and Purified phenolic compounds from the flower of Clitoria ternatea L. (PPCFCTL) should be defined only once in Abstract and text; then the abbreviation should be used.

Response 1: Thanks for your valuable suggestion, we have corrected it.

Point 2: Please check reference #27 (the names should be reversed - now the first names appear)

Response 2: Thanks for your valuable suggestion, we have corrected it.

Reviewer 2 Report

Dear authors.

Thank you for the corrections made and the questions answered. Please convert the photos to black and white before publication and please improve the contrast. Good photography documentation is important in cell culture studies. I think it is worth thinking about a new microscope integrated with a camera. This will allow you to obtain reproducible image quality. I wish you every success. 

Author Response

Response: Thanks for your valuable suggestion, we have corrected it.